# A Review of Sensitivity Enhancement in Interferometer-Based Fiber Sensors

**DOI:** 10.3390/s22072506

**Published:** 2022-03-25

**Authors:** Zengrun Wen, Ziqing Guan, Jingru Dong, Hongxin Li, Yangjian Cai, Song Gao

**Affiliations:** 1Center of Light Manipulations and Applications & Shandong Provincial Key Laboratory of Optics and Photonic Device, School of Physics and Electronics, Shandong Normal University, Jinan 250014, China; wenzengrun@sdnu.edu.cn (Z.W.); yangjiancai@sdnu.edu.cn (Y.C.); 2School of Physics and Electronics, Shandong Normal University, Jinan 250014, China; 202009100224@stu.sdnu.edu.cn (Z.G.); 202009100302@stu.sdnu.edu.cn (J.D.); 202009100204@stu.sdnu.edu.cn (H.L.)

**Keywords:** fiber optics sensors, fiber optics, high-sensitivity fiber sensors, fiber materials

## Abstract

Optical fiber sensors based on an interferometer structure play a significant role in monitoring physical, chemical, and biological parameters in natural environments. However, sensors with high-sensitivity measurement still present their own challenges. This paper deduces and summarizes the methods of sensitivity enhancement in interferometer based fiber optical sensors, including the derivation of the sensing principles, key characteristics, and recently-reported applications.The modal coupling interferometer is taken as an example to derive the five terms related to the sensitivity: (1) the wavelength-dependent difference of phase between two modes/arms $\partial \phi_{d}/\partial \lambda$, (2) the sensor length $L_{w,A}$, (3) refractive index difference between two modes/arms $\Delta n_{eff,A}$, (4) sensing parameter dependent length change α, and (5) sensing parameter dependent refractive index change γ. The research papers in the literature that modulate these terms to enhance the sensing sensitivity are reviewed in the paper.

## 1. Introduction

Optical fiber sensors have become a research hotspot for decades. A good variety of optics fiber sensors have been developed, such as magnetic field sensors [1], ultrasonic detectors [2,3], salinity sensors [4], molecular sensing [5], humidity sensor [6], liquid analyte sensing [7,8], tilt sensors [9], and pressure sensors [10,11]. Different techniques have been proposed for achieving the sensing by long-period fiber gratings [12,13,14,15,16], photonic crystal fibers [13,17,18], fiber Bragg gratings [19,20,21,22], Fabry–Perot interferometers [23,24,25,26,27,28], mode-locked lasers [29], Raman scattering [5], and Stimulated Brillouin scattering [30].

Among manifold fiber-optic sensors, the interferometer-based fiber optic sensor is one of the most important devices since it offers high sensitivity, high resolution, easy operation, and multiplexing capabilities. Although Fabry-Perot interferometry-based sensors can react to changing physical environments through changes as visibility of fringes and finesse of cavity [31], monitoring the wavelength shift of the interference spectrum is a common way for interferometry-based sensors when the surrounding environment changes [32,33,34,35,36]. To date, a variety of reviews aim to categorize fiber optic interferometric sensors according to their operating principles, fabrication methods, and application fields [31,37]. For example, the performances of the modal-interferometer fiber optical sensor have been reviewed based on the standard fibers, polarization maintaining fiber, photonic crystal fiber, and fiber Bragg grating [38]; and a review of the basic sensing platforms implemented using tapered optical is presented exploited to facilitate the development of fiber-optic physical, chemical, and bio-sensors [39]. In this paper, we aim to offer a summary of the interferometer-based fiber-optic sensors including mode-coupling-based sensors, Mach–Zehnder-interferometer-based sensors, Sagnac-interferometer-based sensors, and birefringence-coupling-based sensors. Moreover, the sensing sensitivity is numerically analyzed, which shows that the sensitivity is related to the terms of (1) the wavelength-dependent difference of phase between two modes/arms $\partial \phi_{d}/\partial \lambda$, (2) the sensor length $L_{w,A}$, (3) refractive index difference between two modes/arms $\Delta n_{eff,A}$, (4) sensing parameter dependent length change α, and (5) sensing parameter dependent refractive index change γ. If a high sensitivity is obtained, at least one of these terms should be enhanced. It is likely that this review of the sensitivity enhancement of the interferometer-based fiber-optic sensors is helpful for readers’ understanding of their state-of-the-art applications.

## 2. Sensing Principle of Interferometer-Based Sensors

The interferometer-based sensors monitor the wavelength shift of the interference spectrum when the surrounding environment changes. Then, the key point is to find the relationship between the wavelength shift and the changing parameters. There are many types of interferometers, such as mode-coupling-based sensors, Mach–Zehnder-interferometer-based sensors, Sagnac-interferometer-based sensors, and birefringence-coupling-based sensors. Here, an interferometer with a coupling of two modes is taken as an example to derive the theoretical sensing principle and other techniques can be applied to a similar analysis.

In a mode-coupling inteferometer, let us assume that a fiber sensor structure sustains two main modes (mode *e* and mode *o*). When light is launched into the fiber sensor, the mode *e* and mode *o* are excited and propagate along the sensor. The output pattern is a superposition of the fields of the mode *e* and mode *o* and is given by
$$E=\sqrt{0.5}a_{e}\exp\left[j2\pi n_{eff}^{e}L_{w}/\lambda \right]+\sqrt{0.5}a_{o}\exp\left[j2\pi n_{eff}^{o}L_{w}/\lambda \right].$$

The intensity at the output port is given by
$$I=0.5{\left|a_{e}\right|}^{2}+0.5{\left|a_{o}\right|}^{2}+\left|a_{e}\right|\left|a_{o}\right|\cos\left[\phi_{d}\left(\lambda ,A\right)\right],$$
where $\phi_{d}\left(\lambda ,A\right)=2\pi \Delta n_{eff}L_{w}/\lambda +\theta_{e}-\theta_{o}$, $\Delta n_{eff}=n_{eff}^{e}-n_{eff}^{o}$, $a_{i}$ is the complex amplitude with *i* being *e* or *o*, $\theta_{i}$ is the phase of $a_{i}$, $n_{eff}^{i}$ is the effective refractive index of mode *i*, $L_{w}$ is the length of the sensor, and *A* is the parameter characterizing the change of the surrounding environment.

Troughs are observed in the transmission spectrum when the difference between the phases of mode *e* and mode *o* satisfies the condition $\phi_{d}\left(\lambda ,A\right)=\left(2m+1\right)\pi$, where *m* is an integer. Changes in the surrounding environment *A* from *A* to A+ΔA lead to a change in the phase-difference $\phi_{d}$ given by
(1)$$\Delta \phi_{d}\left(\lambda ,A\right)=\left(\partial \phi_{d}/\partial \lambda \right)\Delta \lambda +\left(\partial \phi_{d}/\partial A\right)\Delta A.$$

The phase-difference at the *m*th trough in the transmission spectrum always satisfies $\phi_{d}\left(\lambda ,A\right)=\left(2m+1\right)\pi$; as the surrounding environment changes from *A* to A+ΔA, the *m*th trough shifts from wavelength $\lambda_{m,A}$ to wavelength $\lambda_{m,A+\Delta A}=\lambda_{m,A}+\Delta \lambda_{m,A}$ such that $\phi_{d}\left(\lambda_{m,A},A\right)=\phi_{d}\left(\lambda_{m,A}+\Delta \lambda_{m,A},A+\Delta A\right)=\left(2m+1\right)\pi$ leading to $\Delta \phi_{d}\left(\lambda ,A\right)=0$. In addition, Equation (Equation 1) becomes
$$\Delta \lambda_{m,A}=-{\left(\partial \phi_{d}/\partial \lambda \right)}^{-1}\left(\partial \phi_{d}/\partial A\right)\Delta A.$$

Due to $\phi_{d}\left(\lambda ,A\right)=2\pi \Delta n_{eff}L_{w}/\lambda +\theta_{e}-\theta_{o}$, the term of $\partial \phi_{d}/\partial A$ becomes $\partial \phi_{d}/\partial A=\left(2\pi /\lambda \right)\left(L_{w,A}\partial \Delta n_{eff,A}/\partial A+\Delta n_{eff,A}\partial L_{w,A}/\partial A\right)$ and then the wavelength-shift $\Delta \lambda_{m,A}$ becomes
(2)$$\Delta \lambda_{m,A}=-{\left(\frac{\partial \phi_{d}}{\partial \lambda }\right)}^{-1}\frac{2\pi }{\lambda }L_{w,A}\Delta n_{eff,A}\left(\alpha +\gamma \right)\Delta A,$$
where $\Delta n_{eff,A}$ is the refractive indices difference between mode *e* and mode *o*, $L_{w,A}$ is the taper length, $\alpha =\left(1/L_{w,A}\right)\partial L_{w,A}/\partial A$, and $\gamma =\left(1/\Delta n_{eff,A}\right)\partial \Delta n_{eff,A}/\partial A$.

To achieve a high sensitivity to the change of the surrounding environment ΔA, the value of $\Delta \lambda_{m,A}$ should be large, which requires that the terms of (1) the wavelength-dependent difference of phase between two modes/arms $\partial \phi_{d}/\partial \lambda$, (2) the sensor length $L_{w,A}$, (3) refractive index difference between two modes/arms $\Delta n_{eff,A}$, (4) sensing parameter dependent length change α, and (5) sensing parameter dependent refractive index change γ should be enhanced. In the next section, the research about sensitivity enhancement utilized the terms of the ${\left(\partial \phi_{d}/\partial \lambda \right)}^{-1}$, $L_{w,A}$, $\Delta n_{eff,A}$, α and γ are reviewed, respectively. Table 1 presents the sensing characteristics of different interferometer-based sensors with different modulated terms.

## 3. Interferometer-Based Sensors

All the terms in Equation (Equation 2) play a role in sensitivity enhancement of interferometer-based sensors. The sensing sensitivity is usually a combined effect of many terms and here we review the most significant one determining the enhanced sensitivity. For example, many sensors based on Mach–Zehnder interferometers utilize the coupling between the fundamental mode and a high-order mode and the high-sensitive sensing performance is achieved by monitoring the shift of the transmission spectrum of the two modes. The high-order mode is usually more sensitive to the surrounding environment change, indicating a large value of γ; the refractive index difference $\Delta n_{eff,A}$ between the fundamental mode and the high-order mode is large, which also leads to a large wavelength shift in the transmission spectrum based on Equation (Equation 2). There are many research papers showing that multiple terms dominate the sensitivity [51,52,53,54,55,56] and various environmental parameters are sensed, such as magnetic field [57] and chemical vapor [58].

Recently, a label-free DNA biosensor based on an exposed core microstructured optical fiber has been demonstrated to utilize a multimode Mach–Zehnder interferometer by splicing a section of tapered exposed core fiber between two single-mode fibers [40]. The binding of biomolecules on the surface of the optical fiber induces the change of the refractive index of the surrounding environment and the tapered design provides the evanescent field with the ability of refractive index detection. Two terms ($\Delta n_{eff,A}$ and γ) are modulated to enhance the measurement sensitivity. Figure 1 shows that, as the refractive index of the external environment increases, the effective refractive index of the high-order mode is more affected by the external refractive index than the fundamental mode, indicating a large value of γ using this structure. Figure 2 presents relationship between the effective refractive index difference ($\Delta n_{eff,A}$) between the fundamental mode and the high-order mode and the external refractive index. The sensor design can be optimized depending on the values of γ and $\Delta n_{eff,A}$ calculated in Figure 1 and Figure 2.

In the next sections, the following five key terms related to the sensitivity enhancement of the interferometer-based fiber sensors are reviewed:Three methods for modulation of ${\left(\partial \phi_{d}/\partial \lambda \right)}^{-1}$;Three methods for modulation of $\Delta n_{eff}$;Modulation of $L_{w,A}$ to enhance the sensor sensitivity;Four methods for modulation of γ;Modulation of α to enhance the sensor sensitivity.

### 3.1. Modulation of ${\left(\partial \phi_{d}/\partial \lambda \right)}^{-1}$ to Enhance the Sensor Sensitivity

#### 3.1.1. Dual-Core Tapers with Large Diameters

A high-sensitivity temperature and strain sensor based on a dual-core As_2_Se_3_-PMMA taper was proposed [41], which modulates the value of $\partial \phi_{d}\left(\lambda \right)/\partial \lambda$ by increasing the As_2_Se_3_ core diameter. The values of ${\left(\partial \phi_{d}/\partial \lambda \right)}^{-1}$ can be calculated as a function of diameter of As_2_Se_3_ core using Comsol Multiphysics with parameters found in [59]. Based on Equation (Equation 2), a large value of ${\left(\partial \phi_{d}/\partial \lambda \right)}^{-1}$ induces a large wavelength shift of the transmission spectrum when the surrounding temperature and imposed longitudinal strain changes, which enhances the measurement sensitivity with the temperature sensitivity of 572 pm/°C and strain sensitivity of −6.23 pm/με.

#### 3.1.2. Effective Group–Velocity Matching Induced by an Antisymmetric Long-Period Grating

Another way to enhance the value of ${\left(\partial \phi_{d}/\partial \lambda \right)}^{-1}$ is based on effective group-velocity matching between the even mode and odd mode in a dual-core taper with an antisymmetric long-period grating [60]. An antisymmetric long-period grating is inscribed by launching optical pulses into a dual-core As_2_Se_3_-PMMA taper, which causes the effective group-velocity matching between the even and odd modes propagating around the central wavelength of the input pulses [61]. The value of $\partial \phi_{d}/\partial \lambda$ tends to 0 near the resonance wavelength of the antisymmetric long-period grating indicating that, near the resonance wavelength, the large value of ${\left(\partial \phi_{d}/\partial \lambda \right)}^{-1}$ leads to a large wavelength shift of the transmission spectrum and then the measurement sensitivity is enhanced.

The sensitivity enhancement based on a dual-core taper with an antisymmetic long-period grating is achieved and temperature measurement sensitivity is of 0.121 nm/°C that is enhanced by a factor of 4.0 due to the large value of ${\left(\partial \phi_{d}/\partial \lambda \right)}^{-1}$ induced by an antisymmetric long-period grating.

#### 3.1.3. Cut-Off Wavelength of High-Order Modes

The value of ${\left(\partial \phi_{d}/\partial \lambda \right)}^{-1}$ can also be modulated near the cut-off wavelength of high-order modes in a optical waveguide. An optical fiber refractive index sensor consisting of an optical microfiber Mach–Zehnder interferometer was demonstrated near the cut-off wavelength of mode $HE_{12}$ [42]. Around the cut-off wavelength, the mode $HE_{12}$ ceases to propagate and only the fundamental mode $HE_{11}$ propagates, leading to a constant value of $\phi_{d}$ and then a large value of ${\left(\partial \phi_{d}/\partial \lambda \right)}^{-1}$. Figure 3 shows the evolution of the transmission spectrum of the microfiber MZI with the variation of refractive index, showing that the refractive index sensitivity around the cut-off wavelength of $HE_{12}$ (up to 44271 nm/RIU) is much higher than that far away from the cut-off wavelength.

### 3.2. Modulation of $\Delta n_{eff}$ to Enhance the Sensor Sensitivity

There are many ways to enhance $\Delta n_{eff}$ in the interferometer-based sensors, such as utilizing the coupling between the fundamental mode and high-order modes, inducing a waveguide with a high refractive index in one arm of an MZI, and using a large refractive index difference between the two polarization axes in polarization-maintaining fibers.

#### 3.2.1. The Coupling between Fundamental Mode and High-Order Modes

A fiber strain sensor was reported by inducing higher-order interference modes using a torsional multimode fiber [62]. As is shown in Figure 4a, the procedure of single mode–twisted-multimode–single mode fiber structure (STMS) is presented, and the twist multimode-fiber region is able to couple more power of light into the cladding and introduce high-order LP modes coupling. Figure 4b presents the transmittance spectrum of SMS and STMS, respectively, which shows that the STMS strucure gives a trough with a higher contrast in the transmission spectrum. The coupling between the fundamental mode and high-order modes has a large $\Delta n_{eff}$, which leads to a high strain sensitivity of −42.5 pm/με based on Equation (Equation 2).

A curvature sensor based on a thin-core fiber Mach–Zehnder interferometer was proposed with the sensor head composed by a short section of thin-core fiber embedded between two single-mode fibers [43]. The high-order fiber cladding mode $LP_{018}$ is excited because of the dramatic structural discontinuity at the splicing point of the single-mode fiber and the thin-core fiber. The Mach–Zehnder interferometer works on the basis of interference between the core mode $LP_{01}$ and the excited cladding mode $LP_{018}$ in the thin-core fiber, and the large refractive index difference $\Delta n_{eff}$ between the fundamental mode $LP_{01}$ and high-order mode $LP_{018}$ achieves a sensitivity of −13.53 nm/m^−1^.

Similarly, a high-sensitivity temperature sensor was also proposed based on a Mach–Zehnder interferometer fabricated by concatenating two microcavities separated by a middle section [63]. High-order cladding modes are excited when light propagates the first microcavity and the interference pattern of the fundamental mode and cladding modes are formed when light propagates the second microcavity. The large refractive index $\Delta n_{eff}$ between the fundamental mode and high-order cladding mode induces a large wavelength shift when the temperature changes in the range of *n* the range of 500–1200 °C with a high sensitivity of 109 pm/°C.

#### 3.2.2. Inducing a Waveguide with a High Refractive Index in One Arm of a MZI

The way of increasing the value of $\Delta n_{eff}$ in a Mach–Zehnder interferometer is usually to form the Mach–Zehnder interferometer using two different waveguides as the two arms, respectively [4,64]. For example, a fiber in-line Mach–Zehnder interferometer fabricated by femtosecond laser micromachining for refractive index measurement was achieved [65]. As is shown in Figure 5, one of the interferometer arms in the air contains a microcavity created by removing part of the fiber core near the core and cladding interface, and the other arm lies in the remaining part of the fiber core, which induces a large effective index difference between the two arms of the MZI to achieve high sensitivity.

#### 3.2.3. Using the Large Refractive Index Difference of the Two Polarization Axes in Polarization-Maintaining Fibers

To obtain a large value of $\Delta n_{eff}$, the large refractive index difference of the two polarization axes in polarization-maintaining fibers can be utilized [66]. A Sagnac-interferometer biosensor was reported based on an exposed core microstructured optical fiber that has noncircular symmetry and thus exhibits birefringence [44]. Figure 6 shows a schematic diagram of the proposed Sagnac interferometer. Light from a broadband source is split into clockwise and counterclockwise arms with two polarization states as it enters the Sagnac loop. The two polarization states experience different optical paths and interfere with each other when the two beams are recombined at the output port. In addition, then the large wavelength shift is achieved by a large refractive index difference $\Delta n_{eff}$ between the two polarization states.

### 3.3. Modulation of $L_{w,A}$ to Enhance the Sensor Sensitivity

Longer sensors have been used for many applications, such as salinity sensing [67]. In addition, it has been demonstrated that the longer interference length of $L_{w,A}$ leads to a larger wavelength shift. A dual-taper-based Mach–Zehnder interferometer was fabricated and applied for refractive index sensing demonstrating that the Mach–Zehnder interferometer with a longer interferometer length has a higher sensitivity [45]. A humidity sensor based on an in-fiber Mach–Zehnder interferometer was proposed, which demonstrated that the high sensitivity is achieved by increasing the length of the $L_{w,A}$ of the Mach–Zehnder interferometer [46].

### 3.4. Modulation of γ to Enhance the Sensor Sensitivity

#### 3.4.1. Specialty Fiber That Is Sensitive to the Surrounding Environment

As shown in Figure 7a, a temperature sensor consisting of a segment of multimode fiber with a polymer cladding spliced between two single-mode fibers was proposed. The packaging material employed in the experiments was an aluminum tube and the structure was bonded to the aluminum tube by using epoxy. The measured temperature sensitivity is 3.195 nm/°C, which is shown in Figure 7b [47]. The high sensitivity of temperature measurement is mainly attributed to the high thermo-optic coefficient (a large value of γ) induced by the polymer cladding. The core material with a large thermal-optic coefficient can also be utilized to enhance γ [68].

#### 3.4.2. Coating Sensitive Materials on the Surface of the Sensors

Depending on the environmental parameter that needs to be sensed, different materials can be coated on the surface of the sensors. For example, a humidity sensor was demonstrated utilizing an interferometer structure that is coated by gelatin [69], gold nanoparticles [70], or a thin film of tin oxide [71] working as sensitive materials to improve the value of γ. In addition, a high-sensitive temperature sensor can be coated with a material with a high thermal-optic coefficient such as a film of polydimethylsiloxane [72] or deionized water [73]. Figure 8 shows the experimental setup of coated polydimethylsiloxane-based MZI temperature sensor with the advantage of the low Young’s modulus, good linear thermal expansion effect, and thermo-optic effect from the polydimethylsiloxane material.

#### 3.4.3. Filling the Air Holes in Photonic Crystal Fibers Using a High Sensitive Solution

A temperature sensor based on a fiber loop with an alcohol-filled photonic crystal fiber was demonstrated [48]. The alcohol has a large thermo-optic coefficient (a large γ), which leads to a large wavelength shift of the interference spectrum of the fiber loop when the surrounding temperature changes. Figure 9a shows the SEM image of the used photonic crystal fiber, and the alcohol is filled in the air holes in the PCF structure. Figure 9b shows the transmission spectra of the alcohol-filled HiBi-PCF FLM at a temperature range of 20 °C to 34 °C and the sensitivity is as high as 6.6 nm/°C, which is 660 and seven times higher than that of an FBG and that of the FLM made of a conventional HiBi fiber.

#### 3.4.4. High-Order Modes

High-order modes are usually more sensitive to the change of the surrounding environment and the excitation of high-order cladding modes can be controlled by changing the waveguide structure such as tapering single-mode fibers and coreless fibers [74]. A fiber refractometer based on a tapered multimode fiber between two single-mode fibers was proposed with a sensitivity better than 1900 nm/RIU [75]. The tapered multimode fiber enables high-order cladding modes, which provides a large overlap with the surrounding solutions. High order modes are more sensitive to the surrounding environment leading to high sensitivity of the refractive index measurement [76].

### 3.5. Modulation of α to Enhance the Sensor Sensitivity

Polymer fibers usually have a large α that is related to the length change when the sensing parameter changes. An ultrasonic sensor based on polymer optical fibers in a Mach–Zehnder interferometer has been proposed [49]. The sensitivities of two Mach–Zehnder interferometric sensors under the same conditions were investigated and compared: One is based on a polymethylmethacrylate fiber and the other based a silica fiber. The sensitivity of the polymer optical fiber is more than 12 times larger than that of the fused silica fiber due to the low Young’s modulus of polymer fibers since polymer fibers have a large deformation under the same strain leading to a large value of α.

Similarly, a high-sensitive temperature and strain measurement has been demonstrated in a microwire with an As_2_Se_3_ core and a PMMA cladding [50]. This structure combines the large thermal-expansion coefficient of the PMMA cladding (α) and the large difference between the refractive-indices of As_2_Se_3_ core and PMMA cladding ($\Delta n_{eff}$) to achieve a temperature sensitivity of −115 pm/°C and strain sensitivity of −4.21 pm/με.

## 4. Discussion

In this review, the methods of sensitivity enhancement are summarized based on the five terms derived by monitoring the wavelength shift of the interferometer-based sensors, and there are a few points that need to be noted. Firstly, wavelength is not the only parameter to monitor in the interferometer-based fiber sensors and there are other parameters, such as intensity change that can be linked to the environmental changes. Secondly, as described in Section 3, all the terms in Equation (Equation 2) play a role in sensitivity enhancement of interferometer-based sensors and sometimes it is very difficult to distinguish which term is the most significant oneFinally, there are limitations for the presented methods: (1) Modulation of the term ${\left(\partial \phi_{d}/\partial \lambda \right)}^{-1}$ always requires changing the waveguide structure or choosing high-order modes, which makes it difficult to achieve; however, once the conditions are met, the sensitivity is much enhanced; (2) Modulation of the term $\Delta n_{eff}$ is often together with the modulation of another term; (3) A larger value of $L_{w,A}$ provides a higher sensitivity, but the modulation of the term $L_{w,A}$ is impractical, since the size of the point sensor should be as small as possible; (4) Modulation of the term γ is a common way for the sensitivity enhancement; as the development of material fabrication technique and fiber fabrication technique, it is easy to find a material that has a large γ to the sensing parameters; (5) Modulation of the term α is mostly associated with the length-related sensing parameters such as temperature and vibration.

## 5. Conclusions

Although interferometer-based fiber optic sensors have been widely applied for various parameter sensing in a variety of applications, there remains a limited knowledge of methods for sensitivity enhancement. This paper reviews various structures and methods that have been used for enhancing the sensitivity, including the derivation of the sensing principles, key characteristics, and recently-reported applications. Five terms related to the sensitivity enhancement of the interferometer-based fiber sensors are reviewed: $\partial \phi_{d}/\partial \lambda$, $L_{w,A}$, $\Delta n_{eff,A}$, α, and γ.

## Figures and Tables

**Figure 1 sensors-22-02506-f001:**
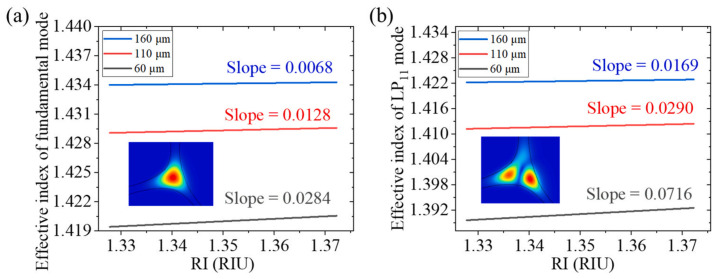
The relationship between external RI and the resulting change in effective mode index under different diameters: (**a**) fundamental mode; (**b**) high-order mode [40]. Reprinted with permission from Ref. [40], Copyright 2022 Elsvier Science and Technology Journals.

**Figure 2 sensors-22-02506-f002:**
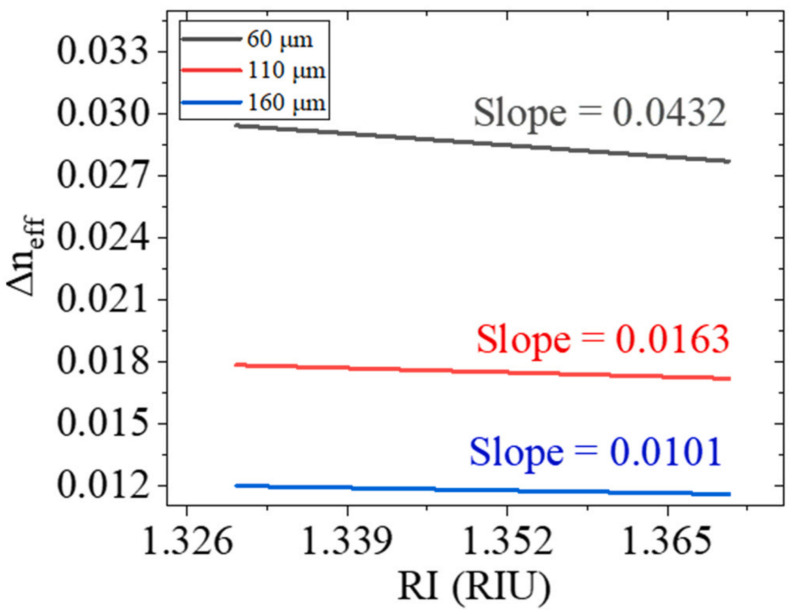
The relationship between the refractive index and $\Delta n_{eff,A}$ (between the fundamental mode and high-order mode) for three fiber diameters [40]. Reprinted with permission from Ref. [40]. Copyright 2022 Elsvier Science and Technology Journals.

**Figure 3 sensors-22-02506-f003:**
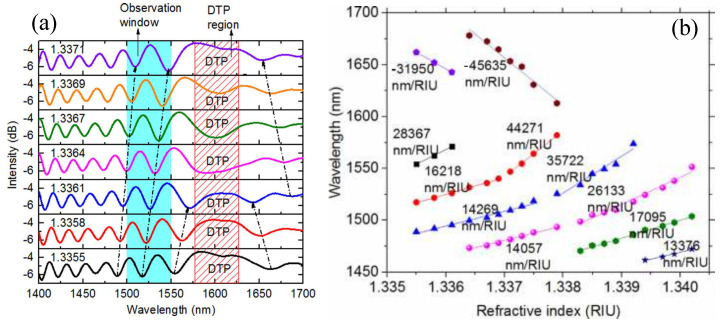
(**a**) Evolution of transmission spectrum of the microfiber MZI with the variation of refractive index; (**b**) RI sensitivities of interference dip around cut-off wavelength of the microfiber MZI in RI range of 1.3355–1.3402. [42]. Reprinted with permission from Ref. [42]. Copyright 2020 Elsvier Science and Technology Journals.

**Figure 4 sensors-22-02506-f004:**
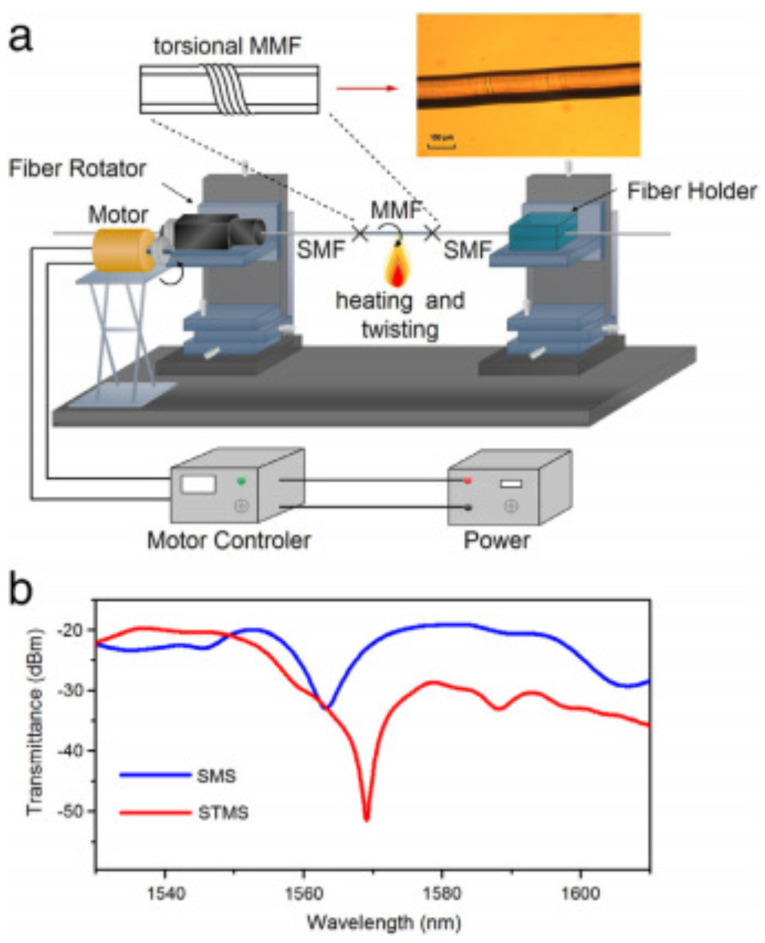
(**a**) Equipment and process of fabricating single mode–twisted-multimode–single mode fiber structure; (**b**) the transmittance spectrum of single mode–multimode–single mode fiber structure (SMS) and single mode–twisted-multimode–single mode fiber structure (STMS), respectively [62]. Reprinted with permission from Ref. [62]. Copyright 2017 Elsvier Science and Technology Journals.

**Figure 5 sensors-22-02506-f005:**
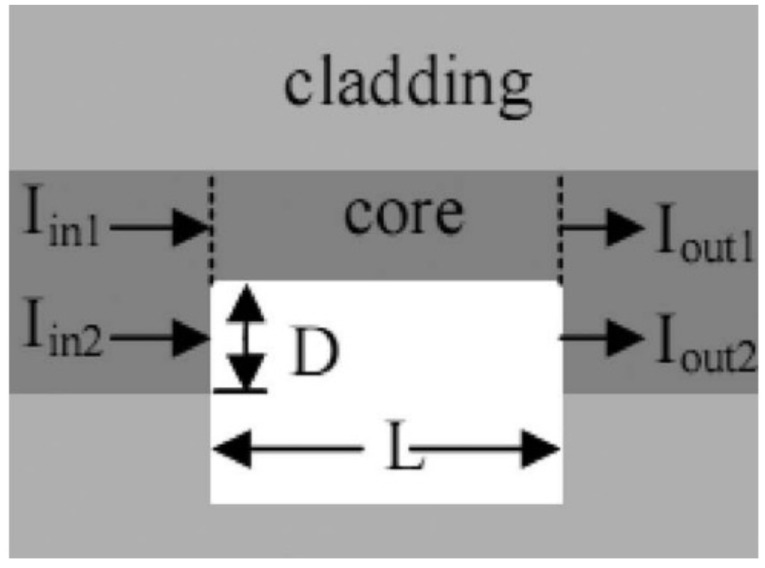
Schematic of the MZI structure fabricated by the use of femtosecond laser pulse irradiation [65]. Reprinted with permission from Ref. [65]. Copyright 2010 Optica Society.

**Figure 6 sensors-22-02506-f006:**
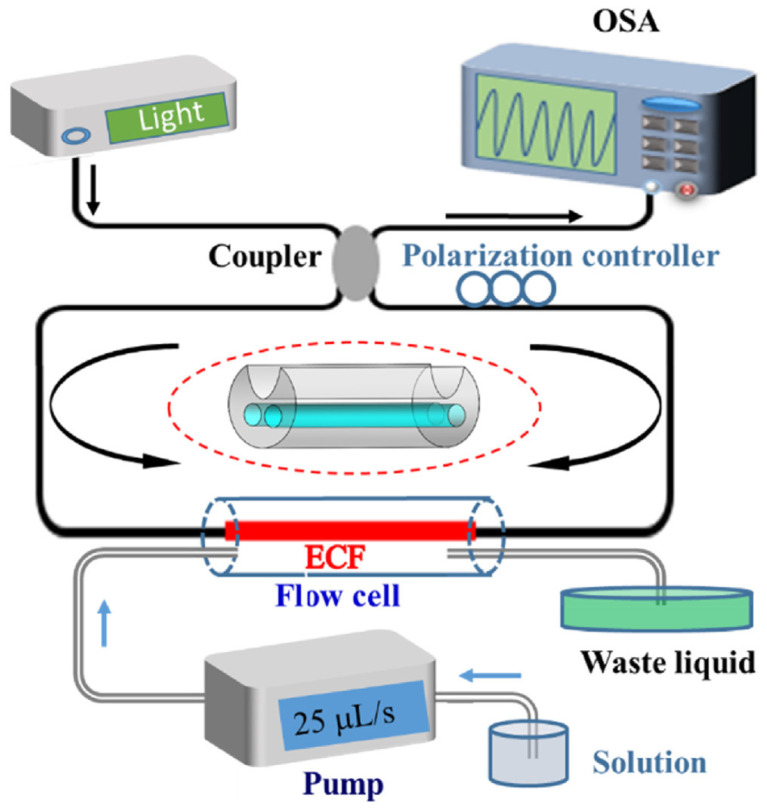
Schematic diagram of the proposed sensing system [44]. Reprinted with permission from Ref. [44]. Copyright 2018 Elsvier Science and Technology Journals.

**Figure 7 sensors-22-02506-f007:**
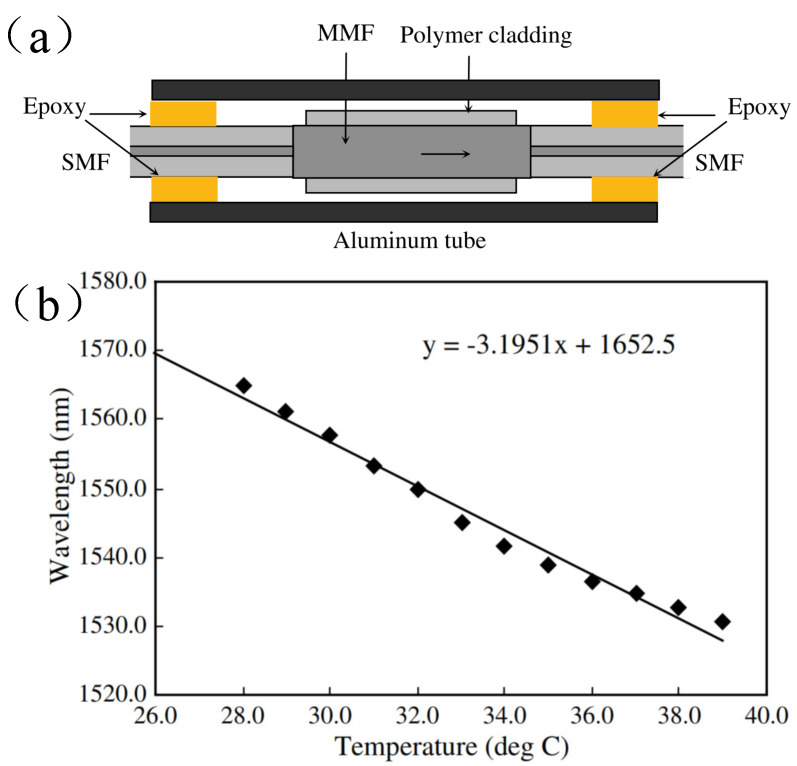
Schematic diagram of the proposed sensing system [47]. Reprinted with permission from Ref. [47]. Copyright 2008 Elsvier Science and Technology Journals.

**Figure 8 sensors-22-02506-f008:**
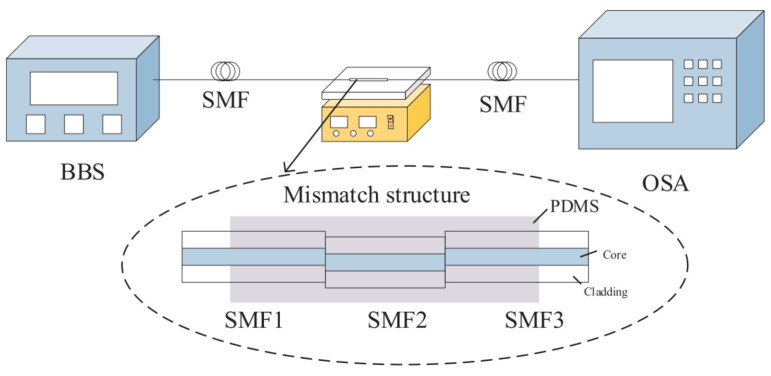
Experimental setup of coated polydimethylsiloxane based MZI temperature sensor [72]. Reprinted with permission from Ref. [72]. Copyright 2019 Elsvier Science and Technology Journals.

**Figure 9 sensors-22-02506-f009:**
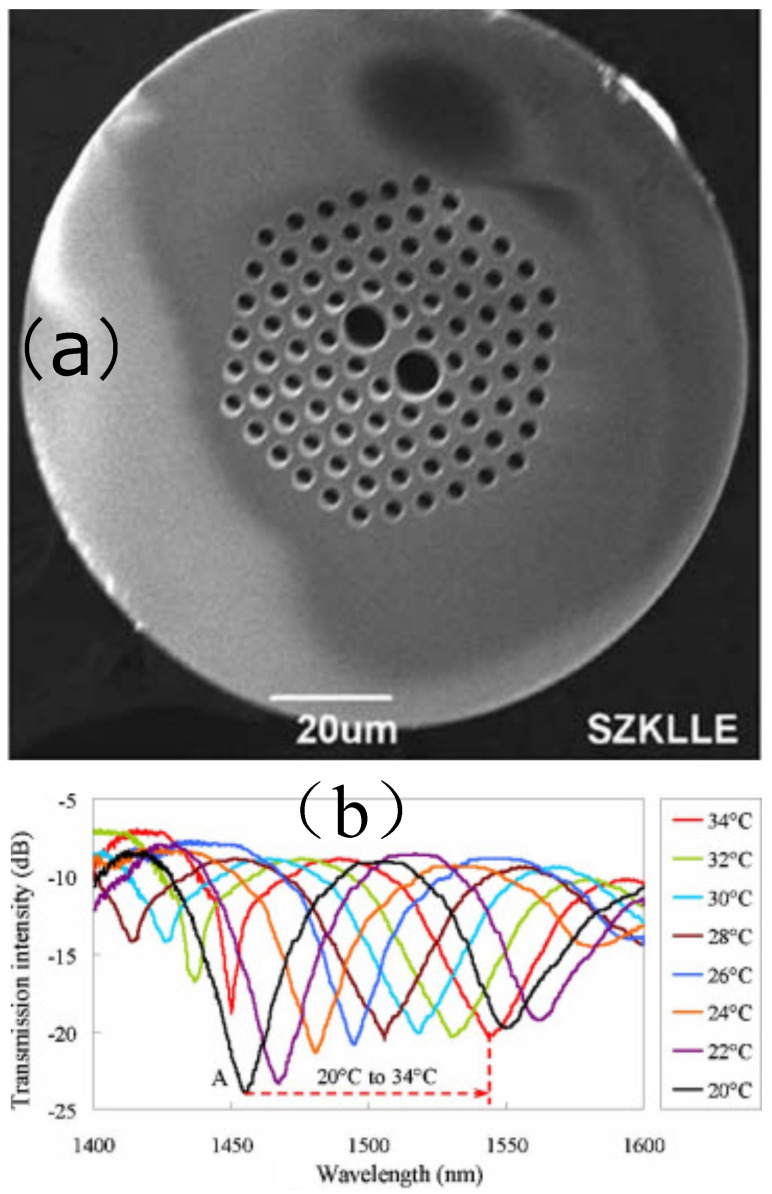
(**a**) SEM of the used photonic crystal fiber; (**b**) transmission spectra of the alcoholfilled HiBi-PCF FLM (**a**) when the temperature increases from 20 °C to 34 °C [48]. Reprinted with permission from Ref. [48]. Copyright 2011 The Optical Society.

**Table 1 sensors-22-02506-t001:** The sensing characteristics of different interferometer-based sensors.

Sensor	Interferometer Type	Modulated Terms	Measurand	Sensitivity	Ref.
Tapered exposed-core fiber	MZI	γ; $\Delta n_{eff,A}$	cDNA	61.8 pm/nM	[40]
Large diameter dual-core taper	Mode Coupling	${\left(\partial \phi_{d}/\partial \lambda \right)}^{-1}$	Temperature; Strain	572 pm/°C; −6.23 pm/με	[41]
Microfibers MZI	MZI	${\left(\partial \phi_{d}/\partial \lambda \right)}^{-1}$	Refractive index	44,271 nm/RIU	[42]
Thin-core fiber MZI	MZI	$\Delta n_{eff}$	Curvature	−13.53 nm/m^−1^	[43]
Exposed core microstructured fiber	Sagnac Interferometer	$\Delta n_{eff}$	Refractive index	−3137 nm/RIU	[44]
Dual taper-based MZI	MZI	$L_{w,A}$	Refractive index	2210.84 nm/RIU	[45]
Arc-induced tapers	MZI	$L_{w,A}$	Humidity	−0.047 nm/%RH	[46]
Polymer fiber	MZI	γ	Temperature	3.195 nm/°C	[47]
Alcohol-filled photonic crystal fiber	Birefringence Coupling	γ	Temperature	6.6 nm/°C	[48]
Polymer optical fibers MZI	MZI	α	Ultrasound	13.1 mrad/kPa	[49]
Dual-core As_2_Se_3_-PMMA taper	Mode Coupling	α; $\Delta n_{eff}$	Temperature; Strain	−115 pm/°C; −4.21 pm/με	[50]

## Data Availability

This paper did not report any data.

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
