# Peer review of "A Review of Sensitivity Enhancement in Interferometer-Based Fiber Sensors"

_sensors, 2022, doi:10.3390/s22072506_

Round 1

Reviewer 1 Report

The submitted work reviewed fiber interferometric sensors and methods for enhancing their sensitivities. The information organization of the paper is well structured for conveying the focus of the review. Although the review is not exhaustive, this work serves as a good starting point for researchers or engineers looking to improve sensitivity of such class of sensors.

1.Comments:

Regarding Line 19: Citation at the end of the sentence contains error. Regarding Line 22: There are examples of interferometric fiber sensors that reacts to changing physical environments through changes as simple as visibility of fringes and finesse of cavity (eg. FP interferometry-based sensors) instead of wavelength shifts.

Therefore, it is not advised to generalize wavelength shifts as the only way to interrogate such sensors. There are published reviews on FPI optical fiber sensors in this “MDPI-Sensors” journal.

2. General comment:

2.1. Each subsection of section 3 was very brief. While the subsections served the purposed of directing reader to at least 1 appropriate work and provide results, author should provide brief discussion on each. As experts in the field, these discussions can help reader gain brief understandings of these works quicker.

2.2. Authors can consider either adding discussion to each enhancement scheme presented in the manuscript or include an overall discussion before conclusion to help reader appreciate the content. This will be helpful if the content of such discussion includes structured way to weigh pros and cons of the methods presented and other important considerations to implement these schemes.

All-in-all, this manuscript do present a good read for researchers first embarking finding solutions for such problems.

Author Response

We have revised our original submission, “A review of sensitivity enhancement in interferometer-based fiber sensors ” (ID#sensors-1598240) in light of the comments from the reviewer. We take this opportunity to thank the reviewer for taking the time to review our manuscript. We have changed the manuscript to address the reviewer’ comments and hope that these changes will meet the reviewer’s requirements.

Reviewer 2 Report

The manuscript presents a review of interferometric fiber sensors with various structures and describes the methods for enhancing their sensitivity. I do not recommend the manuscript for publication due to the following.

- Several reviews on this topic have already been published. They are not even mentioned.

Lee, B.H.; Kim, Y.H. Interferometric fiber optic sensors. Sensors 2012, 12, 2467–2486.

Islam, M.R.; Ali, M.M. Chronology of Fabry-Perot interferometer fiber-optic sensors and their applications: A review. Sensors 2014, 14, 7451–7488.

Nur Hidayah, S.; Hanim, A.R. Modal interferometer structures and splicing techniques of fiber optic sensor. J. Telecomm. Electron. Comp. Eng. 2018, 10, 23–27.

Korposh, S.; James, S.W.; Lee, S.-W.; Tatam, R.P. Tapered optical fibre sensors: Current trends and future perspectives. Sensors 2019, 19, 2294.

It is not clear what is new in the presented review.

- The presented review is far from comprehensive. Many works are not cited. For example:

Kong, Y.; Shu, X. Thin-core fiber taper-based multi-mode interferometer for refractive index sensing. IEEE Sens. J. 2018, 18, 8747.

Bakurov, D.D.; Ivanov, O.V. Control of Excitation of Cladding Modes by Tapering an Insertion of Special Fiber. Sensors 2021, 21, 2498.

Wang, X.A.; Tian, K. High-temperature humidity sensor based on a singlemode-side polished multimode-singlemode fiber structure. J. Lightwave Technol. 2018, 36, 2730–2736.

Vicente, A.; Santano, D. Lossy mode resonance sensors based on nanocoated multimode-coreless-multimode fibre. Sens. Actuators B Chem. 2020, 304, 126955.

- It looks like random papers have been chosen for more detailed description with citation of figures, which contain more details than needed for a review paper.

Author Response

(The authors gave the same response as above.)

Reviewer 3 Report

The paper provides a review of the methods for sensitivity enhancement in fiber sensors. As it is a review, a summary of usable methods are presented. For the reader creates a good background for future investigation. Unfortunately some improvement is needed for clear meaning of the statements.

Chapter 2 needs some more explanation regarding to the parameter description and more details will be valuable.

In table 1 in raw Arc-induced tapers sould be the unit of nm / %RH? (/ is missing?)

In raw 97 is simulation using Comsol mentioned. More parameters including setting of the models, parameters and results would be an advantage.

In text in chapters 3.1.X are the sensitivity values mentioned. Sometimes is the same as in table 1, sometimes different. More discussion about this is needed.

Also some recommendation and author comments on the methods will add very much value.

Author Response

(The authors gave the same response as above.)

Round 2

Reviewer 1 Report

Thank you for considering my comments.